Belief in just deserts regarding individuals infected with COVID-19 in Japan and its associations with demographic factors and infection-related and socio-psychological characteristics: a cross-sectional study

Murakami Michio michio@cider.osaka-u.ac.jp 1
Hiraishi Kai 2
Yamagata Mei 3
Nakanishi Daisuke 4
Miura Asako 1 3
1 Center for Infectious Disease Education and Research, Osaka University , Suita , Japan
2 Faculty of Letters, Keio University , Minatoku , Japan
3 Graduate School of Human Sciences, Osaka University , Suita , Japan
4 The Faculty of Health Sciences, Hiroshima Shudo University , Hiroshima , Japan
Chen Chong
Electronic publication date: 2022 Dec 19
Publication date: 2022
Volume: 10
Electronic Location ID: e14545
Received 2022 Aug 9; Accepted 2022 Nov 18
Copyright: ©2022 Murakami et al.
Copyright year: 2022
Copyright holder: Murakami et al.
License: This is an open access article distributed under the terms of the Creative Commons Attribution License, which permits unrestricted use, distribution, reproduction and adaptation in any medium and for any purpose provided that it is properly attributed. For attribution, the original author(s), title, publication source (PeerJ) and either DOI or URL of the article must be cited.
License URL: https://creativecommons.org/licenses/by/4.0/

Keywords: Belief in just deserts, Belief in a just world, COVID-19, Human rights restrictions, Prejudice, Risk perception

Funding: JSPS KAKENHI 19H01750 The Nippon Foundation - Osaka University Project for Infectious Disease Prevention This work was supported by JSPS KAKENHI Grant Number 19H01750 and “The Nippon Foundation - Osaka University Project for Infectious Disease Prevention.” The funders had no role in study design, data collection and analysis, decision to publish, or preparation of the manuscript.

==============================
Prejudice related to the coronavirus disease 2019 (COVID-19) is a social issue worldwide. A possible psychological factor that promotes prejudice is the belief in just deserts (BJD) regarding individuals infected with COVID-19 (i.e., the belief that the infected individual deserves to be infected). The BJD is based on the belief in immanent justice. It is reportedly higher in Japan than in other countries. Therefore, we conducted a cross-sectional study to investigate the BJD among Japanese individuals and clarify its associations with demographic factors or infection-related and socio-psychological characteristics. To this end, we conducted an online questionnaire survey in Japan from August 7–8, 2020, with 1,207 respondents aged 20–69 years. We performed screening to exclude inappropriate responses. We investigated the association between the BJD and demographic factors such as gender and age. We also investigated the association between the BJD and infection-related and socio-psychological characteristics, including risk perception of COVID-19 infection and human rights restrictions (i.e., the degree of agreement with government restrictions on individuals’ behavior during emergencies). Among the surveyed items, human rights restrictions showed a strong association with BJD, followed by risk perception of COVID-19 infection. Men had a slightly higher BJD than women. Our study is significant in that it is the first to investigate the items associated with the BJD, thereby providing foundational information for revising individual perceptions of justice related to COVID-19 and solving prejudice-related issues.

Introduction

Following the coronavirus disease 2019 (COVID-19) outbreak, discrimination related to COVID-19 infection has become a social issue worldwide (Devakumar et al., 2020; He et al., 2020). In addition to discrimination against infected individuals, stigma and prejudice related to race and occupation have also arisen (Bhanot et al., 2021; Lu et al., 2021). High depressive, anxiety, and post-traumatic stress disorder symptoms have been observed among individuals affected by discrimination (González-Sanguino et al., 2020).

One of the possible psychological factors that promote discrimination, stigma, and prejudice is the belief in just deserts (BJD) regarding individuals infected with COVID-19. The BJD is the belief that the infected person deserves to be infected. It is based on the belief in immanent justice (BIJ), which is one of the subscales of the belief in a just world (BJW) (Maes, 1998). The BJW is the belief that individuals live in a just world where they get what they deserve and deserve what they get (Lerner, 1980). Maes (1998) proposed and scaled the BIJ and the belief in ultimate justice (BUJ) as important sub-dimensions of the BJW. The BIJ represents “the tendency to perceive or see justice in the events that have occurred,” whereas the BUJ represents “the tendency to believe that forthcoming events will settle any injustice that occurs” (Maes, 1998). While the belief is not necessarily bad, the BIJ tends to attribute the cause of harm to the victim (Maes, 1998) and inflicts severe punishment on the perpetrator (Murayama & Miura, 2015). Significantly, since COVID-19 is an infectious disease, infected individuals are liable to infect other individuals. Therefore, understanding the characteristics of BJD regarding individuals infected with COVID-19 is important to address the issues of discrimination, stigma, and prejudice related to COVID-19. The BJD regarding individuals infected with COVID-19 has been included as a relevant item in a series of surveys on exclusionary attitudes toward out-group members and thoughts on following national policies in various countries after the pandemic (OSF Registries, 2020). This item was included because discrimination against individuals infected with COVID-19 has been a big social issue since just after the pandemic (The Japan Times, 2020), and the belief that “the individuals infected with COVID-19 get what they deserve” in Japan is considered to promote excessive self-responsibility and discrimination (Yoshioka & Maeda, 2020). The Japanese government has developed a public relations campaign to eliminate the preconception that the infection is the fault of the infected individuals, with a view toward reduction of discrimination against COVID-19 (Ministry of Justice, 2022). Indeed, a survey conducted in March–April 2020, which included individuals from Japan, the United States, the United Kingdom, Italy, and China, reported a higher BJD score among Japanese individuals compared to individuals from other countries (Miura, Hiraishi & Nakanishi, 2020). This is consistent with another finding that the Japanese are more likely to engage in immanent justice reasoning than Americans (Murayama, Miura & Furutani, 2021). However, to the best of our knowledge, no study has investigated how demographic factors or infection-related and socio-psychological characteristics are associated with the BJD. The investigations of the BJD regarding individuals infected with COVID-19 are expected to provide us with perspectives on discrimination in Japan and insights into individual perceptions of justice related to COVID-19.

Therefore, in this study, we investigated the BJD regarding Japanese individuals infected with COVID-19. As the first article to investigate the BJD with regard to individuals with COVID-19 in detail, we examined the relevant variables in an exploratory manner. In particular, we explored the BJD’s associations with demographic factors or infection-related and socio-psychological characteristics.

Given the relationship between the BJW and demographic factors (Furnham, 2003), we included demographic factors as the relevant variables. As basic information from respondents, we surveyed gender, age, region of residence, the population density of residence, educational background, occupation, marital status, and presence or absence of children. These factors are known to be associated with psychological attitudes toward COVID-19 (e.g., willingness to receive COVID-19 vaccine, risk perception of COVID-19 infection, etc.) (Nomura et al., 2021; Yamagata, Teraguchi & Miura, 2021; Adachi et al., 2022).

In the context of the COVID-19 outbreak and behavioral regulation, we included infection-related and socio-psychological characteristics; specifically, the history of COVID-19 infection, infection rate, risk perception (i.e., perceived prediction of probability that the average Japanese individual will be infected with COVID-19 within the next nine months), and human rights restrictions (i.e., agreeing with the government-imposed restriction on individuals’ behavior during emergencies). We included items about the history of infection, infection rate, and risk perception in the survey, as we assumed that if respondents perceived infection as a common event, it would influence their belief that infected individuals deserve to be infected. As discussed, the BIJ is the tendency to believe that certain events (particularly negative outcomes) are the result of past behavior (negative inputs) (Maes, 1998). At the time of the survey (August 7, 2020), the number of infections in Japan was low and the infections were rare events (see ‘COVID-19 status in Japan up until the time of the survey’ for details). We assumed a negative relationship between the BJD and infection-related items, because infection would be a criterion for how “unjust” a world would be. Furthermore, risk perception of COVID-19 is associated with infection prevention behavior (Dryhurst et al., 2020). It is possible that those who have higher demands for infection prevention behaviors toward others believe that individuals infected with COVID-19 get what they deserve.

We included human rights restrictions in our study because extremely strict social norms (or a low level of tolerance) (Gelfand et al., 2011) are likely to be associated with the BJD. The BIJ is formed and reinforced by punishment experiences (Bennett, 2008), and it tends to take the form of punishment on the perpetrator (Murayama & Miura, 2015). A strong regulatory or punitive orientation toward others is often observed during the COVID-19 outbreak and the resulting behavioral restrictions. This punitive orientation is assumed to be positively associated with the BJD. Indeed, in countries such as Japan and the United States, networks of citizens called “self-restraint police” or “groups of local vigilantes” have sought or enforced strict restrictions on other citizens’ behavior such as to the opening of restaurants and entertainment establishments and long-distance travel (Denyer & Kashiwagi, 2020; Ortiz, 2020; Wright, 2021). Furthermore, authoritative individuals have an increased likelihood of making punitive judgments (Lerner, Goldberg & Tetlock, 1998); therefore, we hypothesized that people who support strict human rights restrictions are more likely to exhibit authoritarian tendencies and attribute the cause of the infection to the infected individuals themselves.

Methods

Ethics

The Osaka University Graduate School of Human Sciences Research Ethics Committee approved our study (Approval No. HB022–007). The survey was completed anonymously. We obtained respondents’ consent before conducting the study. Only the respondents who checked the consent box online were included in the study.

COVID-19 status in Japan up until the time of the survey

The World Health Organization declared COVID-19 as a Public Health Emergency of International Concern on January 30, 2020, and a pandemic on March 11, 2020 (Cucinotta & Vanelli, 2020). In Japan, an official statement was issued to declare a state of emergency in seven prefectures on April 7, 2020, after two voluntary stay-at-home requests on February 26 and March 10, 2020. The state of emergency was extended to include all prefectures on April 16, 2020. Then, it was reduced to eight and five prefectures on May 14, 2020, and May 21, 2020, respectively, before being completely lifted on May 25, 2020. By August 7, 2020 (the start date of the main survey), the total number of confirmed COVID-19 cases in Japan was 45,318 (Ministry of Health Labour and Welfare, 2022).

Study design and participants

Our study design was cross-sectional. We asked a company called LSa to obtain responses via an online questionnaire survey with respondents between the ages of 20 and 69 from a panel provider, Marketing Applications, Inc. LSa does not have its own monitors but acts as an intermediary with a survey company that has monitors. Marketing Applications, Inc. is one of the largest panel providers in Japan with approximately 3.5 million registered panels (as of March 2021), has properly protected personal information, and has earned the Privacy Mark (JIPDEC, 2022). Details on the privacy protection are described elsewhere (Marketing Applications Inc, 2022).

The inclusion and exclusion criteria for the survey were as follows. First, we conducted a screening survey, which included 7,221 respondents, from August 6–7, 2020. In this survey, we set up an instructional manipulation check (Miura & Kobayashi, 2019) to detect inattentive respondents. If the respondents did not answer the questions correctly the first time, we gave them a warning and asked them to read the questions carefully. If they did not answer the questions correctly the second time, we removed them from the survey. After the screening, we found 4,553 inappropriate responses (63.1%). This rate was similar to that reported in a previous study (Miura & Kobayashi, 2015). In the main survey, we included 2,668 respondents by excluding inattentive respondents to enhance the result reliability. Then, we set a target number for each of 10 gender and age groups according to Japan’s population distribution (men: 20s, 8%; 30s, 9%; 40s, 12%; 50s, 11%; 60s, 10%; women: 20s, 8%; 30s, 9%; 40s, 12%; 50s, 11%; 60s, 11%). By using the gender and age information of the monitors registered with Marketing Applications, Inc., we terminated the survey upon reaching the target number. Therefore, the final number of respondents was 1,207. The respondents received points that could be redeemed for products. Whether or not to use the points depends on the monitors’ decision. As described below (see “Survey items”), the BJD consists of two items. The crude results for the item “I think anyone who gets infected with the Coronavirus (COVID-19) got what they deserved,” have already been reported in another study (Miura, Hiraishi & Nakanishi, 2020).

Survey items

Details of questionnaire items and their options are available in the Supplementary File in both Japanese (the original language) and English.

The BJD regarding individuals infected with COVID-19

The BJD comprised two items: “I think anyone who gets infected with the Coronavirus (COVID-19) got what they deserved,” and “If anyone had been infected with the Coronavirus (COVID-19), I think it was their fault.” The former item was used in a previous report (Miura, Hiraishi & Nakanishi, 2020). In this study, the BJD was calculated from the two items to confirm the reliability (see ‘Statistical analysis’). For each item, respondents could choose from various options ranging from “Strongly Disagree (1)” to “Strongly Agree (6)” (i.e., a six-point Likert scale). To ensure content validity, multiple experts in this field (KH, DN, and AM) were consulted to develop the BJD items based on the concept of BIJ. The reliability for the BJD is described in ‘Statistical analysis.’

Demographic factors

We considered gender, age, the region of residence (eight regions), the population density of residence, educational background, occupation, marital status, and presence or absence of children as demographic factors in the survey. For population density, we used the values for the respondents’ prefecture of residence (as of January 1, 2020) (e Stat, 2021; Geospatial Information Authority of Japan, 2020). Further, we asked respondents about their educational background. All items other than educational background were registered in advance by monitors from Marketing Applications, Inc. These items were provided to the authors by LSa from Marketing Applications, Inc. along with the other results of the survey responses.

Infection-related and socio-psychological characteristics

We considered the presence or absence of COVID-19 infection in respondents or respondents’ surroundings (history of infection), COVID-19 infection rate in respondents’ residence, risk perception of COVID-19 infection (Yamagata, Teraguchi & Miura, 2021), and human rights restrictions as infection-related and socio-psychological characteristics. With regard to the infection rate, we considered the total number of confirmed COVID-19 cases per 100,000 individuals in the respondents’ prefecture until or on August 7, 2020 (e Stat, 2021; Ministry of Health Labour and Welfare, 2022). For other items, we asked respondents to respond to the questionnaire. We assessed risk perception by asking respondents about their perception of the probability that the average Japanese individual would be infected with COVID-19 by April 30, 2021; the respondents chose an integer from improbable (0) to certain (100) (i.e., a 101-point scale). We considered six items to measure perceptions regarding human rights restrictions during emergencies:

• “In emergencies, it is better to follow government requests for restrictions on freedom of movement,”

• “In emergencies, speech contrary to government policy should be punished by law,”

• “In emergencies, anyone who goes out against government lockdown policy should be punished by law,”

• “In emergencies, every citizen can autonomously take action to ensure that government policies are respected,”

• “In emergencies, it is better to follow government requests for restrictions on freedom of speech,” and

• “In emergencies, every citizen should watch over to ensure that government policies are respected.” The responses ranged from “Strongly Disagree (1)” to “Strongly Agree (7)” (i.e., a seven-point Likert scale).

To ensure content validity, multiple experts in the field (KH, DN, and AM) were consulted to develop the human rights restriction items based on the context regarding “self-restraint police” after COVID-19. The reliability for the human rights restriction items is described in ‘Statistical analysis’ below.

Statistical analysis

To confirm the reliability of the BJD, we used the Spearman–Brown coefficient (Eisinga, Grotenhuis & Pelzer, 2013; Hulin et al., 2001) and Cronbach’s α. Both were high enough at 0.898, so we used the arithmetic mean of the two items (range: 1–6). The human rights restriction items showed one factor extracted in the factor (maximum likelihood method) and parallel analyses (Hori, 2001), with Cronbach’s α of 0.794. This value was also sufficiently high, so we used the arithmetic mean of the six items (range: 1–7). We used logarithmic values for population density and infection rates.

First, we performed a univariate analysis to examine associations with the BJD; we performed t-tests for two groups, analysis of variance (ANOVA) for three or more groups, and Pearson’s correlation for continuous variables. Next, since we found gender and age to be significantly associated with the BJD, we considered other items with p ≤ 0.05 in the univariate analysis variables in partial correlation analysis by using gender and age as control variables. We created dummy variables for categorical variables. We applied a Bootstrap method (10,000 samples) to estimate a 95% confidence interval (CI).

We used IBM SPSS 28 (IBM, Chicago, IL, USA) in this study.

Results

The arithmetic means (standard deviation; SD) of the single-question items, “I think anyone who gets infected with the Coronavirus (COVID-19) got what they deserved” and “If anyone had been infected with the Coronavirus (COVID-19), I think it was their fault” were 2.46 (1.19) and 2.54 (1.20), respectively. The average value and SD of the two items (BJD) was 2.50 (95% CI [2.44–2.57]) and 1.14, respectively (Table 1).

Table 1 Association between the belief in just deserts (BJD) and demographic factors and infection-related and socio-psychological characteristics: univariate analysis.

	N (%) or Arithmetic mean (SD)	BJD
Arithmetic mean (SD)	Effect size
da, η2b, or rc (95%CI)	p #	
The whole participants	1,207 (100%)	2.50 (1.14)	–	–	
Demographic factors	
Men (ref)	608 (50.4%)	2.58 (1.21)	−0.139 (−0.252–−0.026)a	0.02*	
Women	599 (49.6%)	2.42 (1.05)			
Age	45.9 (13.6)	–	−0.090 (−0.146–−0.034)c	0.002*	
Hokkaido region	53 (4.4%)	2.32 (1.22)	0.004 (0.000–0.007)b	0.67	
Tohoku region	70 (5.8%)	2.47 (1.11)			
Kanto region	434 (36.0%)	2.47 (1.07)			
Chubu region	195 (16.2%)	2.64 (1.27)			
Kinki region	254 (21.0%)	2.47 (1.13)			
Chugoku region	68 (5.6%)	2.49 (1.09)			
Shikoku region	33 (2.7%)	2.48 (1.32)			
Kyushu region	100 (8.3%)	2.57 (1.12)			
log10 population density [persons//km2]	2.95 (0.61)	–	−0.012 (−0.068–0.045)c	0.69	
Elementary/junior high/high school	354 (29.3%)	2.54 (1.17)	0.001 (0.000–0.007)b	0.42	
Junior/technical/professional training college	273 (22.6%)	2.42 (1.05)			
University/graduate school	580 (48.1%)	2.52 (1.15)			
Civil servants/executives/ company employees	497 (41.2%)	2.57 (1.22)	0.003 (0.000–0.011)b	0.18	
Self-employed/free enterprise	99 (8.2%)	2.46 (0.98)			
Full-time homemaker, part-time job, student, other	611 (50.6%)	2.45 (1.09)			
Unmarried individuals (ref)	524 (43.4%)	2.58 (1.20)	−0.124 (−0.237–−0.010)a	0.04*	
Married individuals	683 (56.6%)	2.44 (1.08)			
Absence of children (ref)	621 (51.4%)	2.57 (1.18)	−0.120 (−0.233–−0.007)a	0.04*	
Presence of children	586 (48.6%)	2.43 (1.09)			
Infection-related and socio-psychological characteristics	
History of infection: absence (ref)	1163 (96.4%)	2.50 (1.13)	0.103 (−0.198–0.404)a	0.50	
History of infection: presence	44 (3.6%)	2.61 (1.28)			
log10 infection rate [10−5]	1.43 (0.42)	–	−0.011 (−0.067–0.045)c	0.70	
Risk perception	33.7 (24.1)	–	−0.056 (−0.112–0.000)c	0.05*	
Human rights restrictions	3.35 (1.15)	–	0.329 (0.278–0.379)c	<0.001*	
Notes.

N number of respondents

SD standard deviation

CI confidence interval

# t-test, analysis of variance, Pearson’s correlation.

* p ≤ 0.05.

Univariate analysis showed that among demographic factors, the BJD was significantly lower among women than among men (Table 1). Further, it was lower among married individuals than among unmarried individuals and among those with children than among those without children. It also decreased with increasing age. However, no significant association existed between the region of residence, population density, educational background, or occupation and the BJD. Among infection-related and socio-psychological characteristics, the BJD was negatively associated with risk perception and positively associated with human rights restrictions. Regarding the effect size criteria (—d—>0.20, η2 >0.01, and —r—>0.1), these values had the same effect size level despite the differences in their types (Cohen, 1988). The effect size was relatively large for human rights restrictions (r = 0.329 (95% CI [0.278–0.379])) among demographic factors and infection-related and socio-psychological characteristics.

Partial correlation analysis with gender and age as control variables revealed that —r— was highest for human rights restrictions, followed by risk perception (Fig. 1).

Figure 1 Partial correlations for the belief in just deserts (BJD). (A) Married individuals (ref: unmarried individuals), (B) Presence of children (ref: absence of children), (C) Risk perception, (D) Human rights restrictions.

Control variables: gender and age. CI: confidence interval. Areas of circles are proportional to the number of respondents (N).

Discussion

In this study, we examined the BJD among individuals in Japan and investigated its associations with demographic factors and infection-related and socio-psychological characteristics. To this end, we conducted an online survey in August 2020. We identified the items that demonstrated the strongest association.

The BJD’s arithmetic mean was higher than the lowest value on the scale (1) and lower than the theoretical median (3.5). This was consistent with the finding that the score of immanent justice reasoning is lower than the theoretical median (e.g., Callan et al., 2013). The fact that the BJD in this study was higher than the lowest value on the scale indicated that individuals believed that COVID-19 infected individuals deserved what they got to a small extent.

Among demographic factors and infection-related and socio-psychological characteristics, the item most strongly associated with the BJD was human rights restrictions. This item is assessed based on the degree of agreement with government restrictions on individuals’ behavior during emergencies and indicates the level of adherence to strict social norms. A strong association between social norms and infection prevention measures, such as mask-wearing, was also reported among Japanese populations (Nakayachi et al., 2020).

Further, risk perception of COVID-19 infection was weakly and negatively associated with the BJD. However, no significant associations existed between the infection rate in the respondents’ residence or history of infection (i.e., the presence or absence of infection in respondents or respondents’ surroundings) and the BJD. This indicated that subjective perceptions of infection, rather than objective aspects such as infection rates in the residence or history of infection, were important factors in the association with BJD. One possible interpretation is that a higher risk perception of COVID-19 infection is associated with psychological closeness to the infected individuals, such as viewing the infection as normal. Further, there is a relationship between BJW and psychological distance from the victim (Hafer & Bègue, 2005). Risk perception of COVID-19 infection is reportedly associated with the willingness to be vaccinated (Nomura et al., 2021) and preventive health behaviors such as mask-wearing, hand washing, and physical distancing (Dryhurst et al., 2020). This study provides new insights into the association between risk perception and the BJD.

Among demographic factors, the BJD was significantly associated with gender, age, marital status, and the presence of children in the univariate analysis, whereas its association with the region of residence, population density, educational background, and occupation was non-significant. In particular, the difference in the BJD by gender was relatively large among these items, although their effect sizes were small overall. This finding is consistent with that of a previous meta-analysis (O’Connor et al., 1996) where men had a slightly higher BJW than women (the weighted average effect size d was 0.12). The finding that men have slightly higher BJD than women is probably related to their attitudes toward different groups: a previous survey reported that Japanese men tended to show a higher degree of ethnocentrism than Japanese women after the COVID-19 outbreak (Yamagata, Teraguchi & Miura, 2021). In general, there are gender differences in perceptions of various risks, and cultural worldviews are known to be a mediating variable between the two variables (Kahan et al., 2007). Beliefs, attitudes, and perceptions may not be different specifically due to gender; rather, they can be acquired due to the formation of different roles and subsequent worldviews in society as a result of gender differences.

In this study, we focused on the BJD to better understand how discrimination, stigma, and prejudice related to COVID-19 infection can be addressed. We found that the BJD was most strongly associated with human rights restrictions, even after controlling for gender and age. Our study firstly provided insights on individual perceptions of justice related to COVID-19, as well as foundational knowledge to address COVID-19 infection-related discrimination, stigma, and prejudice in Japan. This finding has important implications during the COVID-19 pandemic for another issue apart from the infectious disease itself—improving people’s wellbeing and revisiting people’s sense of justice. The BJD is considered to create a punitive tendency for individuals infected with COVID-19 because the BIJ can bring severe punishment to the perpetrator by attributing the cause of the harm to the victim (Murayama & Miura, 2015). Thus, the mitigation of BJD is expected to play an important role in reducing the tendency to punish others and improving people’s wellbeing. However, it should be noted that the direction of causality between the BJD and human rights restriction or risk perception is unclear. Therefore, identifying the directionality among these variables through longitudinal or interventional studies will provide foundational knowledge for addressing discrimination and understanding individual perceptions of justice related to COVID-19. In particular, our study revealed a strong association between the BJD regarding infected individuals with COVID-19 and human rights restrictions; thus, it would be useful to examine the direction of causality using a cross lagged panel model in longitudinal studies or to investigate the effect of interventions that weaken the BJD regarding individuals infected with COVID-19 on subsequent human rights restrictions (and vice versa) in interventional studies.

Our study has several limitations. First, since it is a cross-sectional study, it does not allow for causal identification. In particular, it is important to note that reverse causality may exist in the associations between human rights restrictions or risk perception and the BJD. As described above, longitudinal or interventional studies are expected to identify the causality. Second, our study may have been subjected to a selection bias because we conducted an online survey. Although gender and age proportion in our survey was based on the actual distribution in Japan, respondents’ educational level (48.1% belonged to a university or a graduate school) was higher than that of the average Japanese individual (19.8% among 20–69 years old in 2010 (the most recent number available in the national census)) (Statistics Bureau of Japan, 2010). Nevertheless, we found no significant association between educational background and the BJD. Moreover, since we conducted an online survey, respondents had the advantage of receiving points in the form of incentives. This meant that participation was independent of respondents’ interest in the survey topic. Third, our study respondents were aged between 20 and 69 years and lived in Japan. This means that the findings may not apply to other age groups and especially other countries and should be interpreted with caution. Fourth, we obtained a low —r— value between 0.012 and 0.327 from the partial correlation analysis. The —r— value = 0.3 is judged as a medium effect size (Cohen, 1988). This indicates that further studies must be conducted to explore the association between the BJD and other factors that have not been investigated in this study (e.g., cultural worldviews; Kahan et al., 2007).

Conclusions

We investigated the association between the BJD and demographic factors and between the BJD and infection-related and socio-psychological characteristics related to COVID-19-infected individuals in Japan. Our findings were as follows:

• The BJD was higher than the lowest score on the scale, confirming that individuals in Japan believe that infected individuals deserve to be infected to a small extent.

• We found a strong association between human rights restrictions and the BJD, followed by risk perception of COVID-19 infection.

• Men had slightly higher BJD than women.

Our study is significant in that it provides insights on individual perceptions of justice related to COVID-19 and foundational information to help mitigate the social issues of discrimination, stigma, and prejudice related to COVID-19 infection.

Supplemental Information

Supplemental Information 1 Data of each respondent

Click here for additional data file.

We would like to thank Editage for English language editing. We are also grateful for helpful discussions: Dr. Yang Li (Nagoya University), Dr. Nobuhiro Mifune (Kochi University of Technology), and Dr. Andrea Ortolani (Rikkyo University).

Additional Information and Declarations

Competing Interests

Author Contributions

Human Ethics

Data Availability

The authors declare there are no competing interests.

Michio Murakami conceived and designed the experiments, analyzed the data, prepared figures and/or tables, authored or reviewed drafts of the article, and approved the final draft.

Kai Hiraishi conceived and designed the experiments, performed the experiments, authored or reviewed drafts of the article, and approved the final draft.

Mei Yamagata conceived and designed the experiments, performed the experiments, authored or reviewed drafts of the article, and approved the final draft.

Daisuke Nakanishi conceived and designed the experiments, performed the experiments, authored or reviewed drafts of the article, and approved the final draft.

Asako Miura conceived and designed the experiments, performed the experiments, authored or reviewed drafts of the article, and approved the final draft.

The following information was supplied relating to ethical approvals (i.e., approving body and any reference numbers):

The Osaka University Graduate School of Human Sciences Research Ethics Committee approved our study (Approval No. HB022–007).

The following information was supplied regarding data availability:

The raw measurements are available in the Supplementary File.

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
