# Peer review of "Belief in just deserts regarding individuals infected with COVID-19 in Japan and its associations with demographic factors and infection-related and socio-psychological characteristics: a cross-sectional study"

_PeerJ, doi:10.7717/peerj.14545_

## Round 0.1 · original submission · Major Revisions

In order to show the identified associations are genuine, can the authors provide a scatterplot for the partial correlations?

Reviewer 1 ·

Basic reporting

• Some bold claims are made in the abstract with no supportive citations. I advise including supportive citations
• I would like to know more about the Miura et al. paper cited in the introduction. What was the BJD measure related to? Covid-19? This seems to be the only citation about BJD so needs greater introduction
• I am not convinced on the rationale for the inclusion of the correlates of BJD – it seems this is quite an exploratory study and this should be stated. Please then include greater detail as to why correlates were included in the introduction.
• Subheadings for the different materials in the materials subsection would be useful
• The citation for the BJD items is missing from the survey items subsection
• It is typical practice to describe findings that do not reach statistical significance “non-significant”, rather than insignificant (line 227)
• I did not understand the claim on line 237 “public sharing of a high level of tolerance may help alleviate BJD” – what do you mean by public sharing? A high level of tolerance of what? I do not think we can make causal claims of alleviating BJD form this study as no experimental design was implemented

Experimental design

• No clear hypotheses or research question was posed at the end of the introduction
• The description of the pandemic in the method section does not seem appropriately placed and I wonder if this should instead be moved to a separate section?
• This study seems exploratory and there is not a good rationale as to why measures have been included.

Validity of the findings

• It is not surprising that BJD mean was lower than the scale mean. Please see the work on Immanent Justice reasoning by Mitch Callan, which shows endorsement of JR is generally low and below the scale midpoint. This does not mean few people believe covid-19 infected individuals deserved what they got, instead it shows that people agreed with this sentiment to a small extent. The comparison should be with the lowest value on the scale perhaps than the midpoint?
• On line 215 you cannot claim this correlation is causal in either way – there maybe a third variable responsible for the relationship here
• I am not sure how seeing COVID-19 as common helps mitigate BJD (line 220)?
• Can you provide more detail on your claim about the gender difference and previous research on lines 231-233? Instead of assuming a probable connection here, perhaps you could propose a future experimental study to investigate such claims
• I have an issue in that this study is proposed as investigating discrimination and prejudice and stigma in relation to COVID-19. Instead, this is an exploratory study considering a number of variables around covid and demographics and how they relate to perceptions of covid-19 being deserved. The authors could argue that participants perceiving covid-19 as deserved does not result in prejudice/discrimination but instead gives us insight into individuals perceptions of justice relating to Covid-19
• The limitations section is generic and surface level with little supporting citations

Additional comments

• I have read and acknowledge the author notes provided
• Raw data from the authors is provided
• No identifiable data is included, but there is always a risk participants could be identified by the pattern of demographic data provided
• The authors confirm ethical approval has been sought

·

Basic reporting

The subject matter is intriguing, and the examples and supporting evidence are illustrative and engaging.
There are a lot of formatting errors which needs to be reviewed and has been commented on the article.

Experimental design

Hypothesis is missing in the method section.
There is sufficient data presented in the results and discussed afterwards. However, substantive editing is
needed to enhance clarity.

Validity of the findings

Overall, the research quality is good. The study presents an interesting objective and is designed to address and meet the aims of the study.

Additional comments

The document is adequately written. I have made a number of changes and suggestions in the document that will improve the general level of language. As mentioned above, the document is written in a clear and concise manner.

---

## Round 0.2 · Minor Revisions

Please clarifying several methodological issues and enhance the rationale as well as implications of the study, as suggested by the reviewer.

Reviewer 1 ·

Basic reporting

• On Page 4 the authors refer to BIJ/BUJ as “subscales of BJW”. I Am not sure this is accurate as there are many different scales of BJW and there is no citation to determine which one the authors are referring to. Instead I think what the authors are trying to say is that BUJ/BIJ are strategies/reactions people may employ when confronted with an injustice to restore a belief in a just world

• I think discussion of the exploratory variables and why they were chosen is needed in the introduction. I did not find the text in the methods section wholly substantiated – greater support is needed as to why variables were included

o On page 11 “. We included items about the 181 history of infection, infection rate, and risk perception in the survey because we considered that if 182 respondents perceived infection as a common event, this would impact their thinking that 183 infected individuals deserve to be infected” – can the authors explain what kind of impact is being alluded to here and why?

o On page 11 “We included human rights restrictions in our study because 187 extremely strict social norms (or a low level of tolerance) (Gelfand et al., 2011) are likely to be 188 associated with the BJD.” – why is this association likely?

• “p” on page 15 should be lower case

Experimental design

• See previous comment about justification of the inclusion of correlates for BJD being needed

• Further details of how the demographic variables and exploratory variables were measured is required in the method section. There is not enough detail in the materials section for someone to replicate this study

• Were the two BJD items averaged and combined? This was not clear in the results/method. Please can the authors explain how they treated the data for all variables?

Validity of the findings

• On page 20 the authors claim the results provide insight on how to address covid-19 discrimination and has important implications for the pandemic to improve people’s wellbeing – what are these implications and insights specifically?

---

## Round 0.3 · accepted · Accept

Thank you for making the revisions.